# IL-23 drives differentiation of peripheral γδ17 T cells from adult bone marrow-derived precursors

Pedro H Papotto[1], Natacha Gonçalves-Sousa[1], Nina Schmolka[1], Andrea Iseppon[2], Sofia Mensurado[1], Brigitta Stockinger[2] (ID), Julie C Ribot[1] & Bruno Silva-Santos[1,3,*] (ID)

## Abstract

Pro-inflammatory interleukin (IL)-17-producing γδ (γδ17) T cells are thought to develop exclusively in the thymus during fetal/perinatal life, as adult bone marrow precursors fail to generate γδ17 T cells under homeostatic conditions. Here, we employ a model of experimental autoimmune encephalomyelitis (EAE) in which hematopoiesis is reset by bone marrow transplantation and demonstrate unequivocally that Vγ4+ γδ17 T cells can develop *de novo* in draining lymph nodes in response to innate stimuli. *In vitro*, γδ T cells from IL-17 fate-mapping reporter mice that had never activated the *Il17* locus acquire IL-17 expression upon stimulation with IL-1β and IL-23. Furthermore, IL-23R (but not IL-1R1) deficiency severely compromises the induction of γδ17 T cells in EAE, demonstrating the key role of IL-23 in the process. Finally, we show, in a composite model involving transfers of both adult bone marrow and neonatal thymocytes, that induced γδ17 T cells make up a substantial fraction of the total IL-17-producing Vγ4+ T-cell pool upon inflammation, which attests the relevance of this novel pathway of peripheral γδ17 T-cell differentiation.

**Keywords** experimental autoimmune encephalomyelitis; IL-17; IL-23; T-cell differentiation; γδ T cells

**Subject Categories** Development & Differentiation; Immunology

## Introduction

Interleukin (IL)-17A (IL-17 herein) is a major promoter of antimicrobial peptide production and neutrophil mobilization, which likely accounts for its conservation across evolution of the vertebrate immune system [1]. On the other hand, the contributions of IL-17 to inflammatory and autoimmune diseases make it a hot target for current and upcoming immunotherapeutic strategies [2].

While CD4+ αβ T cells are certainly the better known producers of IL-17, thus defining the "T helper 17" ($T_H17$) cell lineage [3–5],

they are often preceded and outnumbered at earlier stages of immune responses by γδ T cells [6]. These can indeed mount very rapid IL-17-based responses that drive neutrophil recruitment and control microbial load, as documented in multiple infection settings: *Listeria monocytogenes* in the liver [7]; *Escherichia coli* in the peritoneal cavity [8]; *Bordetella pertussis* in the lung [9]; *Mycobacterium bovis*-BCG in the skin [10]; and *Candida albicans* and *Pseudomonas aeruginosa* in the eye [11], among others (reviewed in Ref. 12). On the other hand, IL-17-producing γδ (γδ17) T cells can promote pathology upon infiltration and accumulation in target tissues. This has been demonstrated in mouse models of diseases such as arthritis [13], colitis [14], uveitis [15], type 1 diabetes (T1D) [16], psoriasis [17–19], and multiple sclerosis [20–22].

γδ17 T cells are also major sources of IL-17 in steady-state conditions [23], likely due to their "developmental pre-programming" in the thymus [24]. Thus, we and others have shown that mouse γδ thymocytes can acquire the capacity to produce IL-17, which associates with the upregulation of CCR6 and the loss of CD27 expression [25,26]. Importantly, the development of γδ17 T cells is believed to be restricted to fetal/perinatal life, as transplantation of adult bone marrow, or induction of Rag1 activity after birth, failed to generate γδ17 T cells [27]. According to this model, steady-state γδ17 T cells are only generated in fetal and neonatal thymus, persisting thereafter as self-renewing and long-lived cells in the thymus and in peripheral organs [27,28], where they can engage in immune responses. Whether γδ T cells derived from adult bone marrow precursors can be induced to express IL-17 in peripheral lymphoid organs under inflammatory conditions still remains unresolved. Indeed, since a substantial fraction of γδ T cells exit the adult thymus as functionally immature ("naïve") T cells, they could differentiate into IL-17 producers upon activation, alike conventional αβ $T_H17$ cells. While this has been shown for a very small (~0.4%) population of γδ T cells whose TCR recognizes the algae protein phycoerythrin (PE) [28,29], it remains unknown whether (and to what extent) such peripheral differentiation occurs in pathophysiological settings. To address this important question, we turned here to the experimental autoimmune encephalomyelitis (EAE) mouse model of multiple sclerosis.

γδ T cells significantly accumulate during the acute phase of EAE [30]; most of these cells bear a Vγ4+ TCR and make IL-17 [22,31].

1 Instituto de Medicina Molecular, Faculdade de Medicina, Universidade de Lisboa, Lisboa, Portugal
2 The Francis Crick Institute, London, UK
3 Instituto Gulbenkian de Ciência, Oeiras, Portugal
*Corresponding author. Tel: +351 21 799 94 66; Fax: +351 21 798 51 42; E-mail: bssantos@medicina.ulisboa.pt

Moreover, contrary to CD4$^+$ T cells, γδ T cells in the inflamed spinal cord remain stable IL-17 producers, as evaluated in a reporter mouse strain designed to fate-map cells that have activated IL-17 production [23]. Such γδ17 T-cell responses depend on the innate cytokines IL-1β and IL-23 [22], which are essential for the induction of EAE [32–34]. The early production of IL-17 by γδ17 T cells was shown to establish an amplification loop that sustains IL-17 production by CD4 + T$_H$17 cells [22]. Most importantly, TCRδ$^{-/-}$ [20–22], like IL-17$^{-/-}$ mice [35], develop attenuated EAE pathology with a delayed onset.

While EAE clearly constitutes an appropriate model to address peripheral γδ17 T-cell differentiation under inflammatory conditions, there is a major confounding factor—the sizeable "natural", that is, thymic-derived γδ17 T-cell pool established in steady-state secondary lymphoid organs since birth. To overcome this problem, we have here induced EAE after resetting hematopoiesis through lethal irradiation followed by bone marrow transplantation. Since adult bone marrow precursors cannot generate thymic γδ17 T cells [27], the transplanted mice are devoid of thymic-derived peripheral γδ17 T cells before EAE induction. This allowed us to unequivocally demonstrate the differentiation of γδ17 T cells from "naïve" γδ T cells in draining lymph nodes in response to inflammatory IL-23 signals.

## Results and Discussion

### Peripheral differentiation of γδ17 T cells upon EAE inflammation

We established bone marrow chimeras (BMCs) using a congenic marker (Thy1.1/Thy1.2) to distinguish donor and host hematopoietic cells and TCRδ$^{-/-}$ recipients, to guarantee the absence of any host γδ T cells that might resist the irradiation protocol (Fig 1A). As expected [27], after 8 weeks of reconstitution, γδ T cells lacked IL-17 but expressed IFN-γ in peripheral organs (Fig 1B; Fig EV1). EAE was induced by injection of myelin oligodendrocyte glycoprotein (MOG) peptide, complete Freund's adjuvant (CFA) and pertussis toxin, as widely established [22]. The BMCs developed severe pathology, comparable to unmanipulated C57Bl/6 mice, with slightly delayed onset (Fig 1C). When we analyzed the BMCs at the peak of disease

(day 14 post-induction; p.i.), we found striking proportions of IL-17$^+$ γδ T cells in the brain, lymph nodes, and spleen, in stark contrast with naïve BMCs (Fig 1B and D). As expected in EAE [22], these γδ17 T cells expressed almost exclusively Vγ4$^+$ TCRs (Fig 1E). Importantly, they also expressed the master transcription factor RORγt, but not T-bet (Fig 1F), the cytokine receptor IL-1R1 (Fig 1G) and the surface molecule CD44 (Fig 1H). These data demonstrate that *bona fide* γδ17 T cells can differentiate in the periphery under inflammatory conditions.

### MOG and TLR-independent peripheral γδ17 T-cell differentiation in lymph nodes

Next, we investigated the generation site of the induced γδ17 T cells in EAE by sacrificing the animals at an early time point (day 7 p.i.), before the appearance of the first clinical signs of the disease. We examined lymphoid organs, the target tissue, and other non-lymphoid tissues implicated in the generation of encephalitogenic cells [36,37] and found γδ17 T cells mainly in the draining lymph nodes (Fig 2A and B), where they actively proliferated, as shown by Ki67 staining (Fig 2C). In some mice, we detected small frequencies of γδ17 T cells also in the cervical lymph nodes (cLN), spleen, and lungs (Fig 2D). While it is possible that these cells can differentiate outside the immunization area (due to propagation of inflammatory signals), they could, alternatively, be recirculating to get licensed to enter the CNS [37]. Importantly, these cells were not found in the brain, which still did not show an inflammatory infiltrate at this time point, nor in the lamina propria, mesenteric lymph nodes (mLN), or thymus (Fig 2A and D).

Given that the EAE induction protocol comprises both myelin-specific antigen (MOG peptide) and innate stimuli derived from CFA and pertussis toxin (PTx), we next administered (subcutaneously) different combinations of the adjuvants in the absence of MOG peptide (Fig 3A). BMCs immunized with CFA plus PTx showed substantial pools of γδ17 T cells in draining lymph nodes, in stark contrast to IFA plus PTx or CFA alone (Fig 3A and B; Fig EV2A). Since the peripheral generation of γδ17 T cells did not require myelin-specific antigens but rather CFA and PTx, we hypothesized that innate cytokine stimuli, rather than recognition of

---

**Figure 1.   Peripheral differentiation of γδ17 T cells upon EAE inflammation.**

A   Experimental setup: bone marrow chimeras (BMCs) were generated by injecting total bone marrow cells from wild-type (WT) Thy1.1$^+$ donor mice into TCRδ$^{-/-}$ (Thy1.2$^+$) hosts. After 8 weeks, these BMCs were immunized s.c. in both flanks with 125 μg of MOG$^{(35–55)}$ peptide emulsified in CFA solution; additionally, BMCs were given 200 ng of PTx i.v. on days 0 and 2 p.i. for additional adjuvant effect. Mice were sacrificed at day 14 p.i., and brain, draining lymph nodes (dLN), cervical lymph nodes (cLN), and spleen were harvested.

B   Flow cytometry analysis of intracellular IL-17A and IFN-γ expression in Thy1.1$^+$CD3$^+$TCRδ$^+$ cells isolated from naïve or EAE-induced Thy1.1:TCRδ$^{-/-}$ BMCs.

C   Mice were observed daily and scored for clinical signs of EAE.

D   Frequencies of IL-17A$^+$ cells within the Thy1.1$^+$CD3$^+$TCRδ$^+$ population in the different organs analyzed. Each symbol represents an individual BMC.

E   Flow cytometry analysis of TCR-Vγ4 and TCR-Vγ1 expression in Thy1.1$^+$CD3$^+$TCRδ$^+$IL-17A$^+$ cells isolated from EAE-induced Thy1.1:TCRδ$^{-/-}$ BMC.

F   Flow cytometry analysis of intracellular RORγt (top panel) and T-bet (bottom panel) expression in Thy1.1$^+$CD3$^+$TCRδ$^+$IL-17A$^+$ cells isolated from the spleen of EAE-induced Thy1.1:TCRδ$^{-/-}$ BMC.

G   Flow cytometry analysis of IL-1RI expression in IL-17A$^+$ (blue), IFN-γ$^+$ (red) or IL-17A$^-$IFN-γ$^-$ (purple) cells within Thy1.1$^+$CD3$^+$TCRδ$^+$Vγ4$^+$ cells isolated from the spleen of EAE-induced Thy1.1:TCRδ$^{-/-}$ BMC. FMO refers to Fluorescence Minus One (FMO) controls (without anti-IL-1R1 antibody) on the same cell population.

H   Flow cytometry analysis of CD44 and IL-17A expression in Thy1.1$^+$CD3$^+$TCRδ$^+$ cells isolated from the spleen of EAE-induced Thy1.1:TCRδ$^{-/-}$ BMC.

Data information: (B–D) "naïve" refers to non-immunized BMCs. (C, D) Data pooled from two independent experiments ($n$ = 4–10 mice per group). (E–H) Data representative of at least two independent experiments. Each symbol represents an individual BMC. (C–H) Error bars represent mean ± SD. (D) *$P < 0.05$ **$P < 0.01$ (Mann–Whitney $U$-test). (G) *$P < 0.05$ (nonparametric one-way ANOVA, Kruskal–Wallis test).

*Mycobacterium tuberculosis* products through Toll-like receptors, drove *de novo* γδ17 T-cell differentiation. Consistent with this, MyD88$^{-/-}$:TCRδ$^{-/-}$ (donor:host) BMCs (immunized with CFA plus PTx) were perfectly capable of inducing γδ17 T cells in draining lymph nodes (Fig 3C and D; Fig EV2B). Additionally, agonists of TLR2/3/4/9 all failed to elicit peripheral γδ17 T-cell differentiation in Thy1.1:TCRδ$^{-/-}$ BMCs (Fig EV2C and D). Altogether, our data indicate that cell-intrinsic TLR signaling is not required for peripheral γδ17 T-cell differentiation. These data beckoned the dissection

of additional signals responsible for the induction of IL-17 expression in uncommitted γδ T cells.

### IL-23-dependent peripheral γδ17 T-cell differentiation

In order to obtain a reliable source of uncommitted γδ T cells, we employed an IL-17 fate-mapping reporter mouse line where eYFP expression permanently marks the activation of the *Il17* locus [23]. We cultured highly purified (> 99%) eYFP(−) γδ T cells with

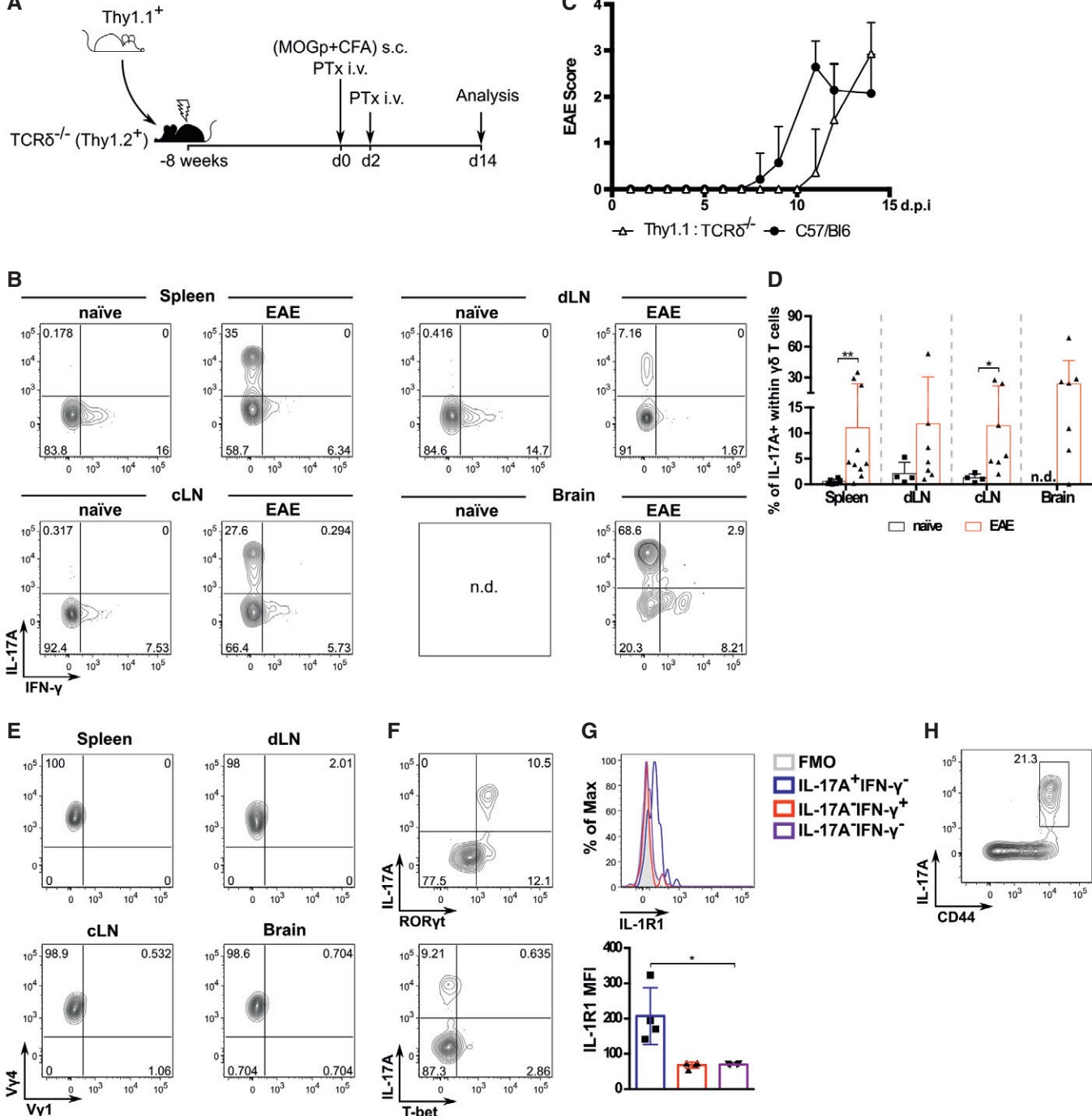

**Figure 1.**

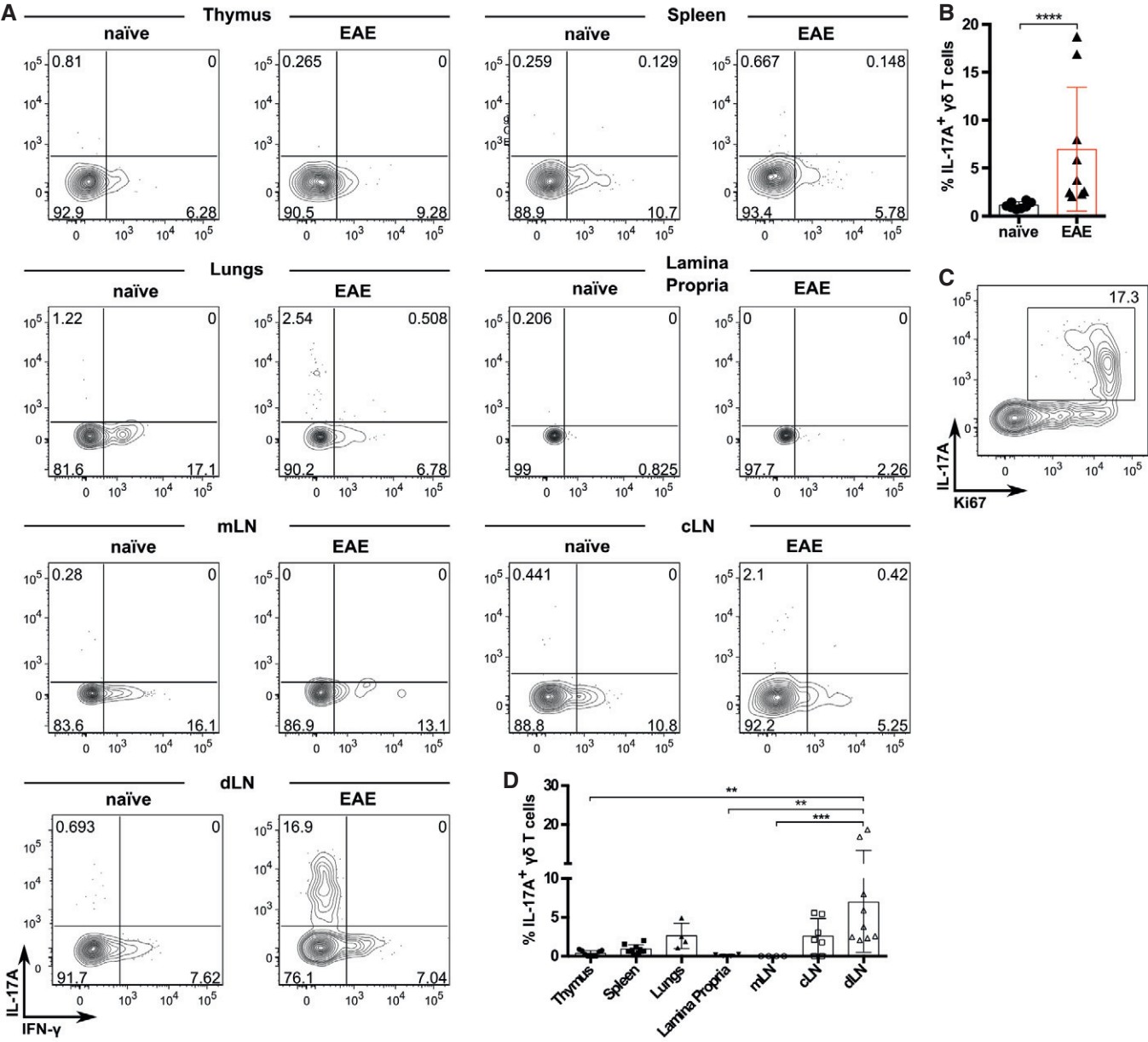

**Figure 2. Peripheral γδ17 T-cell differentiation occurs in draining lymph nodes.**

A  Flow cytometry analysis of intracellular IL-17A and IFN-γ expression in Thy1.1⁺CD3⁺TCRδ⁺ cells isolated from naïve or EAE-induced Thy1.1:TCRδ⁻/⁻ BMCs (n = 5–6 mice per group), established as in Fig 1A.

B  Frequencies of IL-17A⁺ cells within the Thy1.1⁺CD3⁺TCRδ⁺ population in the draining LN of the BMCs in (A).

C  Flow cytometry analysis of intracellular IL-17A and Ki67 expression in Thy1.1⁺CD3⁺TCRδ⁺ cells isolated from EAE-induced Thy1.1:TCRδ⁻/⁻ BMCs (as in A).

D  Frequencies of IL-17A⁺ cells within the Thy1.1⁺CD3⁺TCRδ⁺ population in the organs of the BMCs depicted in (A).

Data information: (A–D) Error bars represent mean ± SD. Data pooled from two independent experiments. Each symbol represents an individual BMC. (A, B) "Naïve" refers to non-immunized BMCs. (A–D) n = 4–9 mice per group. (B) ****P < 0.0001 (Mann–Whitney U-test). (D) **P < 0.01 ***P < 0.001 (nonparametric one-way ANOVA, Kruskal–Wallis test).

various activation/differentiation cocktails (Fig 4A). IL-1β and IL-23 were found to be sufficient to elicit *de novo* γδ17 T-cell generation (Fig 4A and B). *In vitro* stimulation with IL-1β and IL-23 (but not TGF-β or IL-6) had been shown to trigger abundant IL-17 secretion by peripheral CD27- CCR6⁺ γδ T cells [22,26,38,39], but since these cells contained thymic-derived γδ17 T cells, it was not possible to distinguish between expansion of pre-differentiated versus induction

of γδ17 T cells. In our *in vitro* system, although TCR stimulation was not essential, it synergized with these cytokines to greatly enhance the frequency of eYFP⁺ cells (Fig 4A and B). Unexpectedly, addition of IL-6 and TGF-β decreased the mean fluorescence intensity (MFI) of eYFP (Fig 4C).

As our data (Fig 3C and D) argued against a non-redundant role for MyD88-dependent IL-1β/IL-1R1 signaling, we next investigated

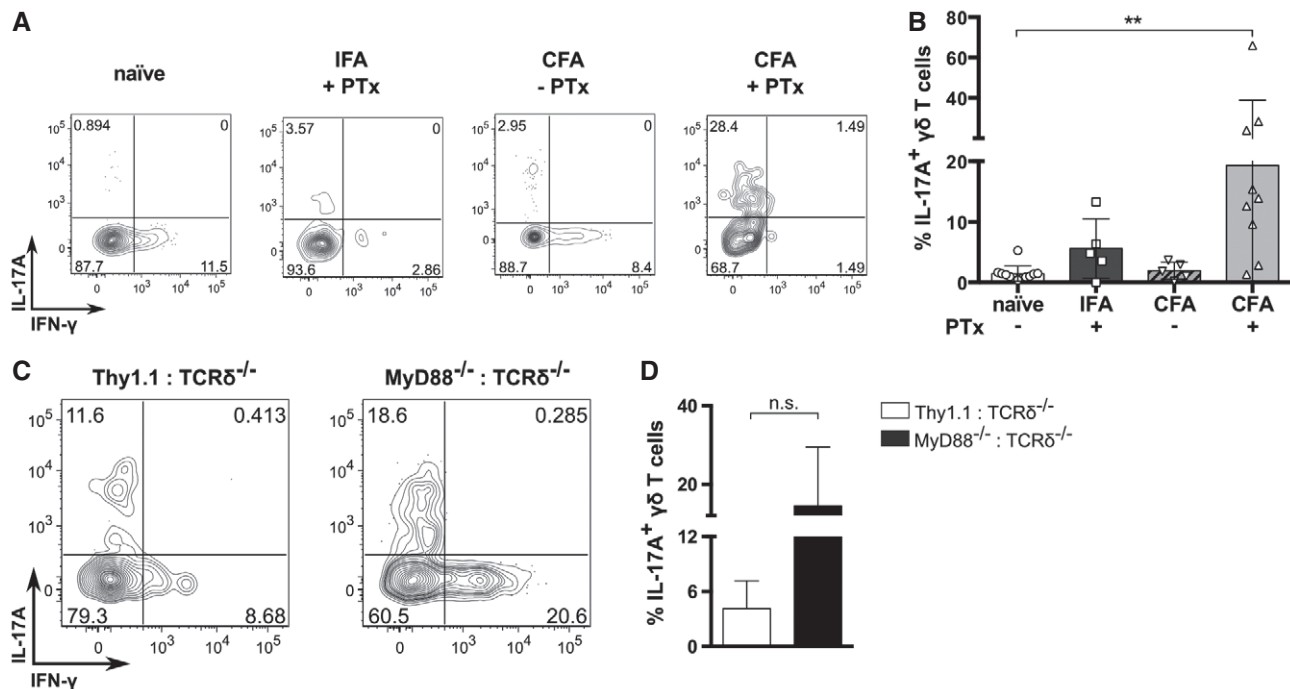

**Figure 3. Peripheral γδ17 T-cell differentiation does not require myelin antigens or cell-intrinsic TLR recognition.**

A    Flow cytometry analysis of intracellular IL-17A and IFN-γ expression in Thy1.1$^+$CD3$^+$TCRδ$^+$ cells isolated from the dLN of Thy1.1:TCRδ$^{-/-}$ BMCs injected subcutaneously with IFA or CFA followed or not by PTx administration.
B    Frequencies of Thy1.1$^+$CD3$^+$TCRδ$^+$IL-17A$^+$ cells within the Thy1.1$^+$CD3$^+$TCRδ$^+$ population in the dLN of the BMCs in (A).
C, D    Thy1.1:TCRδ$^{-/-}$ or MyD88$^{-/-}$:TCRδ$^{-/-}$ BMCs were injected subcutaneously with CFA and given 200 ng of PTx i.v. on days 0 and 2 p.i. for additional adjuvant effect. (C) Flow cytometry analysis of intracellular IL-17A and IFN-γ expression in CD3$^+$TCRδ$^+$ cells isolated at day 7 p.i. Data are representative of two independent experiments. (D) Frequencies of IL-17A$^+$ cells within the CD3$^+$TCRδ$^+$ population in the dLN.

Data information: (A–D) Error bars represent mean ± SD. n.s., not significant. Data pooled from two independent experiments. Each symbol represents an individual BMC. (A) "Naïve" refers to non-immunized BMCs. (A, B) $n$ = 4–10 mice per group; (C, D) $n$ = 6–7 mice per group. (B) **$P$ < 0.01 (nonparametric one-way ANOVA, Kruskal–Wallis test). (D) Mann–Whitney $U$-test.

whether either MyD88-independent IL-1β/IL-1R1 or IL-23/IL-23R signals drove peripheral γδ17 T-cell differentiation *in vivo*. For this, we generated mixed BMC using Thy1.1$^+$ and IL-23R$^{-/-}$ or IL-1R1$^{-/-}$ as donor cells (in a 1:1 ratio), and after 8 weeks immunized them with CFA plus PTx (Fig 4D). As expected, γδ17 T cells

were found in the draining lymph nodes of IL-23R$^{-/-}$ mixed BMCs, but not in their naïve counterparts (Fig 4E and F). Importantly, the vast majority were of Thy1.1 (IL-23R$^{+/+}$) origin (Fig 4G). Moreover, we observed a marked shift in the Thy1.1:IL-23R$^{-/-}$ ratio among total γδ T cells (Fig EV3A and B), which further attests the

**Figure 4. IL-23 drives peripheral γδ17 T-cell differentiation.**

A–C    CD3$^+$TCRδ$^+$eYFP$^-$ cells were FACS-sorted from the peripheral LN and spleen of *Il17a$^{Cre}$R26R$^{eYFP}$* mice and cultured *in vitro* for 72 h in the presence of IL-1β (10 ng/ml), IL-23 (10 ng/ml), IL-6 (10 ng/ml), TGF-β (10 ng/ml), and plate-bound anti-CD3 mAb (10 μg/ml) combined as shown in (A). All conditions also included IL-7 and IL-21 (10 ng/ml each), except condition I, which contained IL-7 (10 ng/ml) only. Data pooled from two independent experiments ($n$ = 7 mice per experiment). (A) Flow cytometry analysis of eYFP expression in CD3$^+$TCRδ$^+$ cells after 72 h under the conditions depicted. Data are representative of two independent experiments. (B) Frequency and (C) mean fluorescence intensity (MFI) of eYFP$^+$ in CD3$^+$TCRδ$^+$ cells (as in A).
D    WT (Thy1.1$^+$) and IL-23R$^{-/-}$ (Thy1.2$^+$) or IL-1R1$^{-/-}$ (Thy1.2$^+$) bone marrow total cells were mixed at 1:1 ratio to reconstitute lethally irradiated TCRδ$^{-/-}$ hosts. After 8 weeks, mice were injected subcutaneously with CFA and given 200 ng of PTx i.v. on days 0 and 2 p.i. for additional adjuvant effect. "Naïve" refers to non-immunized BMCs.
E    Flow cytometry analysis of intracellular IL-17A and IFN-γ expression in CD3$^+$TCRδ$^+$ cells isolated at day 7 p.i. from the dLN of the Thy1.1:IL-23R$^{-/-}$ mixed BMCs (D).
F    Frequencies of IL-17A$^+$ cells within the CD3$^+$TCRδ$^+$ population in the dLN of naïve (white bar) or CFA-immunized (gray bar) Thy1.1:IL-23R$^{-/-}$ mixed BMCs (as in D).
G    Flow cytometry analysis and frequencies of IL-23R$^{+/+}$ (Thy1.1$^+$Thy1.2$^-$; white bar) and IL-23R$^{-/-}$ (Thy1.1$^-$Thy1.2$^+$; black bar) within CD3$^+$TCRδ$^+$IL-17A$^+$ cells from the dLN of CFA-immunized Thy1.1:IL-23R$^{-/-}$ mixed BMCs (as in D).
H    Flow cytometry analysis of intracellular IL-17A and IFN-γ expression in CD3$^+$TCRδ$^+$ cells isolated at day 7 p.i. from the dLN of the Thy1.1:IL-1R1$^{-/-}$ mixed BMCs (D).
I    Frequencies of IL-17A$^+$ cells within the CD3$^+$TCRδ$^+$ population in the dLN of naïve (white bar) or CFA-immunized (gray bar) Thy1.1:IL-1R1$^{-/-}$ mixed BMCs (as in D).
J    Flow cytometry analysis and frequencies of IL-1R1$^{+/+}$ (Thy1.1$^+$Thy1.2$^-$; white bar) and IL-1R1$^{-/-}$ (Thy1.1$^-$Thy1.2$^+$; black bar) within CD3$^+$TCRδ$^+$IL-17A$^+$ cells from the dLN of CFA-immunized Thy1.1:IL-1R1$^{-/-}$ mixed BMCs (as in D).

Data information: (A–J) Each symbol represents an individual BMC. Error bars represent mean ± SD. (E–G) Data pooled from two independent experiments ($n$ = 3–8 mice per group). (H, J) $n$ = 4–5 mice per group. (F, G, I, J). *$P$ < 0.05, ***$P$ < 0.001 (Mann–Whitney $U$-test).

impact of the CFA-induced and IL-23-dependent γδ T-cell response. As for Thy1.1:IL-1R1$^{-/-}$ mixed BMCs, while they harbored γδ17 T cells after immunization (Fig 4H and I) and displayed a shift in Thy1.1:IL-1R1$^{-/-}$ ratio (Fig EV3C and D), they contained a substantial fraction of γδ17 T cells derived from IL-1R1$^{-/-}$ progenitors (Fig 4J). These data collectively suggest that IL-23R (rather than IL-1R1) signaling is the key orchestrator of peripheral γδ17 T-cell differentiation *in vivo*.

## Induced γδ17 T cells make a large contribution to the total γδ17 T-cell pool in EAE

Finally, we aimed to establish whether peripheral γδ17 T-cell differentiation would occur in the presence of "natural" (thymic-derived) γδ17 T cells—and, if so, to determine the relative contributions of the two pools in EAE. To answer these questions, we transplanted neonatal thymocytes (expressing both Thy1.1 and

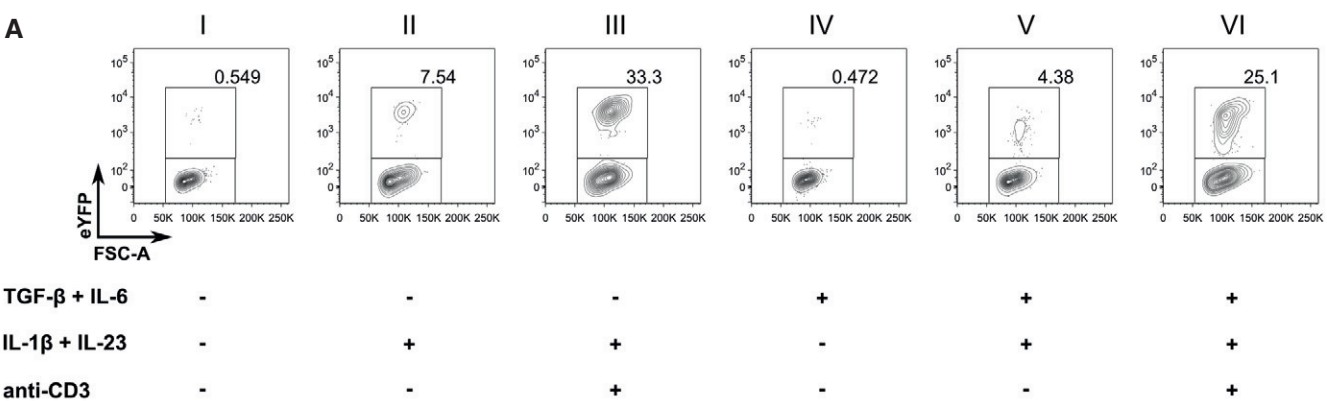

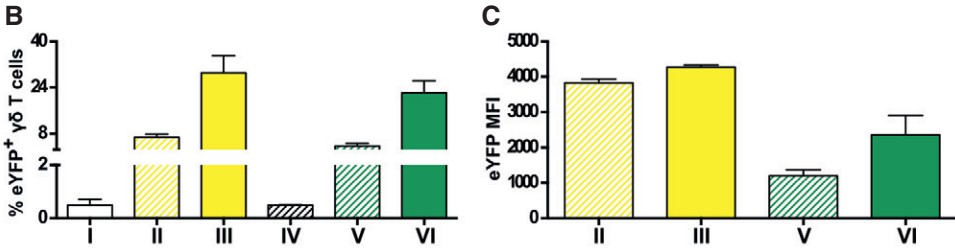

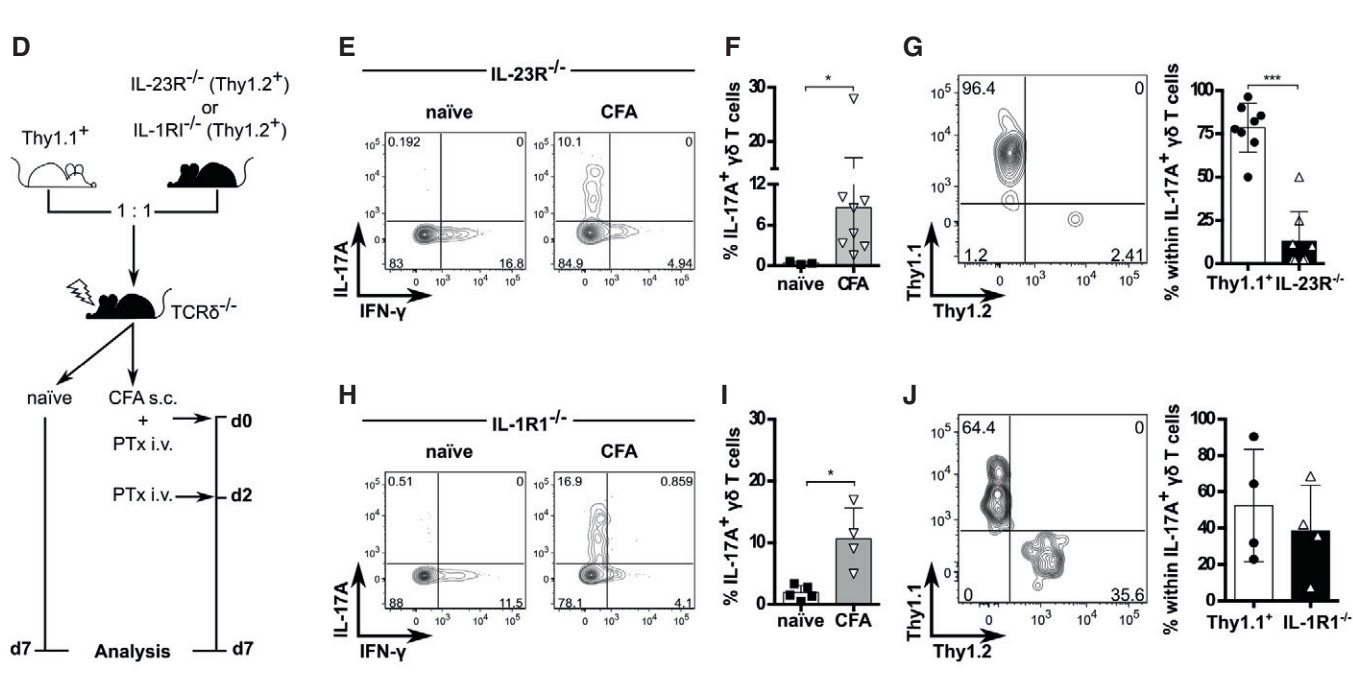

Figure 4.

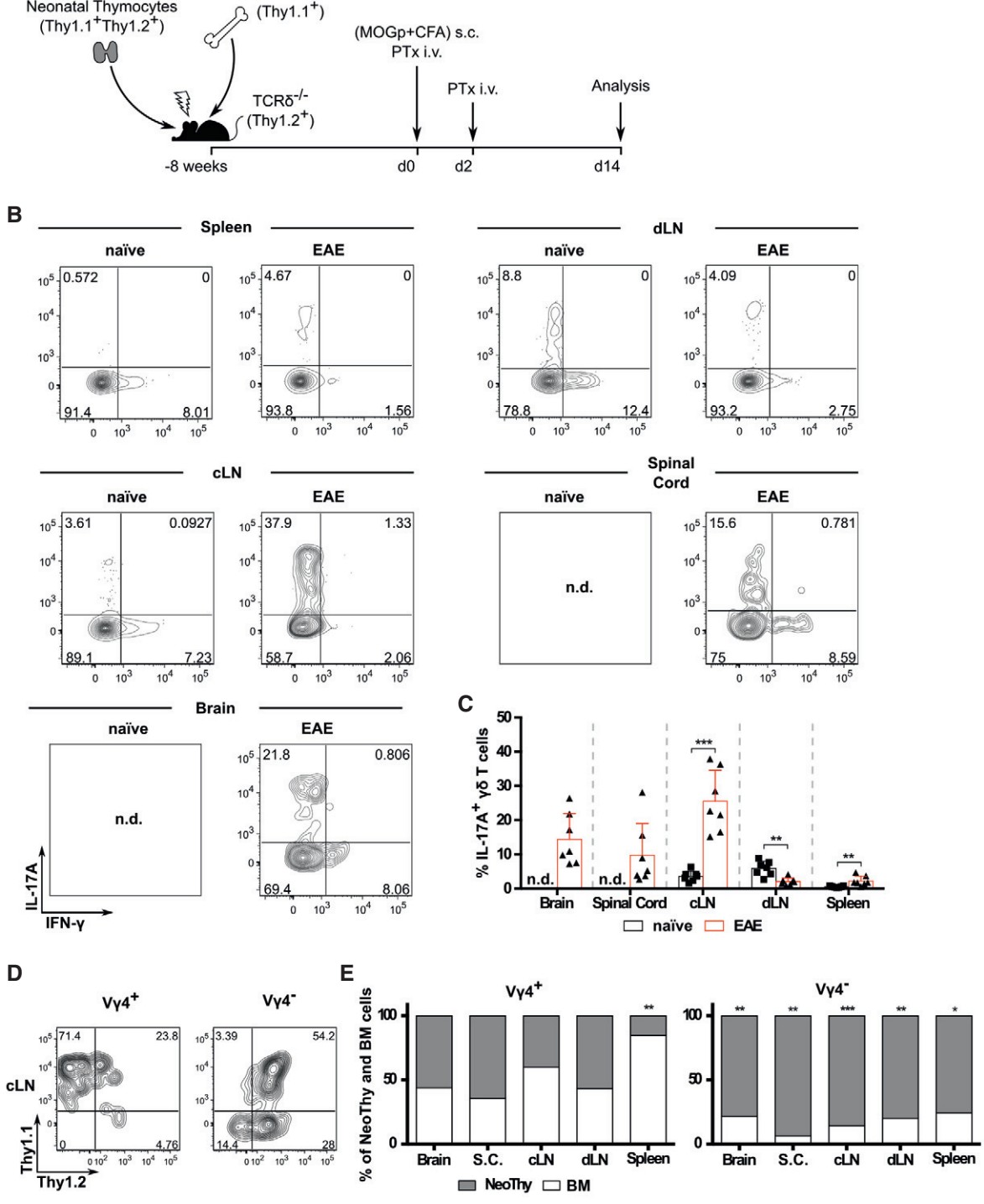

**Figure 5.  Induced γδ17 T cells make a large contribution to the total γδ17 T-cell pool in EAE.**

A    Neonatal thymocytes (NeoThy; Thy1.1+Thy1.2+) and bone marrow cells (BM; Thy1.1+Thy1.2−) were injected into lethally irradiated TCRδ−/− hosts (Thy1.1−Thy1.2+). After 8 weeks, these NeoThy+BM chimeras were immunized s.c. in both flanks with 125 μg of MOG(35–55) peptide emulsified in CFA solution; additionally, BMCs were given 200 ng of PTx i.v. on days 0 and 2 p.i. for additional adjuvant effect. Mice were sacrificed on day 14 p.i., at the peak of the disease, and brain, spinal cord, dLN, cLN, and spleen were harvested.

B    Flow cytometry analysis of intracellular IL-17A and IFN-γ expression in CD3+TCRδ+ cells isolated from naïve or EAE-induced NeoThy+BM chimeras.

C    Frequencies of IL-17A+ cells within the CD3+TCRδ+ population in the different organs analyzed from naïve (black bar) and EAE-immunized (red bar) NeoThy+BM chimeras.

D, E    Flow cytometry analysis (D) and frequencies (E) of NeoThy (Thy1.1+Thy1.2+)- versus BM (Thy1.1+Thy1.2−)-derived cells within CD3+TCRδ+IL-17A+Vγ4+CD44hi (left panels) and CD3+TCRδ+IL-17A+Vγ4−CD44hi (right panels) cells.

Data information: (A–E) Each symbol represents an individual BMC. Error bars represent mean ± SD. Data pooled from two independent experiments (*n* = 7 mice per group). (C, E) *P < 0.05, **P < 0.01, ***P < 0.001 (Mann–Whitney *U*-test)

Thy1.2) and bone marrow cells (Thy1.1$^+$) into TCRδ$^{-/-}$ mice (Fig 5A). As expected [17,27], we could observe "natural" γδ17 T cells of neonatal thymic origin in the lymph nodes of naïve mice (Fig 5B; Fig EV4A). However, upon EAE induction (Fig EV4B), γδ17 T cells were found also in the brain and spinal cord (Fig 5B and C). Of note, these mice presented increased frequencies of γδ17 T cells in cervical LN and spleen, but decreased in the draining LN (Fig 5B and C), probably due to their migration to the central nervous system (CNS). Of interest, the dominant γδ17 T-cell subset in this model switched from Vγ1$^-$Vγ4$^-$ to Vγ4$^+$ cells (Fig EV4C and D). Critically, around half of the Vγ4$^+$ γδ17 T cells in the lymph nodes and CNS during EAE derived from adult bone marrow precursors, whereas Vγ4$^-$ γδ17 T cells were mainly of neonatal thymic origin (Fig 5D and E). These data clearly demonstrate that peripheral γδ17 T-cell differentiation accounts for a large fraction of the total Vγ4$^+$ γδ17 T-cell pool in EAE.

In summary, our study identifies a peripheral pathway of differentiation of *bona fide* RORγt$^+$ γδ17 T cells derived from adult bone marrow precursors, which occurs mainly in the draining lymph nodes upon inflammation, including EAE. We further demonstrate that this pathway does not depend on specific myelin antigens but rather on innate stimuli, as those contained in CFA and PTx. By combining *in vitro* and *in vivo* approaches and gene-targeted mice, we ascribe a key role to IL-23 in the *de novo* differentiation of peripheral γδ17 T cells.

Interestingly, the differentiation of thymic versus peripheral γδ17 T cells seemingly relies on distinct cytokines. Thus, IL-23 is dispensable for thymic γδ17 T-cell development [40], which is instead promoted by TGF-β1 and IL-7 [40–42] and inhibited by IL-15 [43]. By contrast, peripheral induction of γδ17 T cells relies on IL-23 but not TGF-β1 or IL-6 (which in fact decrease IL-17 levels).

Even more complex is the role of the TCR in γδ17 T-cell differentiation. Although TCR engagement synergized with IL-1β plus IL-23 stimulation (to enhance γδ17 T-cell induction), it was *per se* not required for acquisition of IL-17 expression by cells that had never activated the *Il17* locus before. This is consistent with the overall impact of the TCR on peripheral γδ17 T-cell responses [22]; but also with its dispensable role in the development of "natural" Vγ4$^+$ γδ17 T cells in the thymus [20,44]. Interestingly, strong TCR signals promote the development of thymic-derived Vγ6$^+$ γδ17 T cells, which are the other main subset of γδ17 T cells. Whereas in EAE the main responsive γδ17 T-cell subset is Vγ4$^+$, other inflammatory diseases (also) engage Vγ6$^+$ γδ17 T cells. That is the notable case of imiquimod-induced psoriasiform inflammation in the dermis, where Vγ4$^+$ and Vγ6$^+$ γδ17 T cells are differentially involved [17,18]. Therefore, it will be interesting to investigate the peripheral differentiation of Vγ4$^+$ versus Vγ6$^+$ γδ17 T cells in this model.

In a previous study, a discrete population (~0.4%) of γδ T cells was shown to recognize the algae antigen PE via the TCR and differentiate into IL-17 producers [29], although the fetal versus adult origin of the "naive" precursors was not addressed. Most interestingly, cognate PE interactions were shown to upregulate IL-23R (as well as IL-1R1) expression, suggesting that TCR signals were required to "license" the cells to respond to IL-23 (and IL-1β) [29]. We therefore suggest that the mechanisms of peripheral γδ17 T-cell differentiation converge on IL-23R signaling. Consistent with this, γδ T cells constitutively expressing IL-23R are known to be the first cells to respond to IL-23 during EAE development [21]. Moreover,

IL-23 was shown to greatly enhance IL-17 production by γδ T cells triggered by stimulation with TLR2 and Dectin-1 ligands *in vitro* [45]. The presence of pathogen-associated molecular patterns of *M. tuberculosis* and subsequent IL-23 production may thus underlie the expansion of the γδ17 T-cell pool in response to CFA in our and previous studies [45,46]. However, the need for PTx signals in our model points to different requirements and/or thresholds of response to innate cytokines in peripheral γδ17 T cells compared to their thymic counterparts. Thus, PTx is likely required to maximize the innate stimuli provided by CFA (but not IFA) through the production of innate cytokines, as previously described [47,48].

Peripheral γδ17 T-cell differentiation is especially relevant in humans, where γδ thymocytes are functionally immature [49] and require activation under inflammatory conditions to produce IL-17 [50], with IL-23 playing a major role in the process [51]. This reinforces the rational for targeting the IL-23/IL-17 axis in autoimmune diseases, which has already produced remarkable results in psoriasis and shows great potential in multiple sclerosis [52].

## Materials and Methods

### Mice

All mice used were adults 6–18 weeks of age. C57BL/6J.Thy1.1 and C57BL/6J.MyD88$^{-/-}$ (hereafter referred as Thy1.1$^+$ and MyD88$^{-/-}$, respectively) mice were obtained from Instituto Gulbenkian de Ciências (Oeiras, Portugal), the latter with permission from Dr. Shizuo Akira (Osaka University, Osaka, Japan). C57BL/6J.TCRδ$^{-/-}$ and C57Bl/6J.IL-1R1$^{-/-}$ mice were purchased from The Jackson Laboratory. C57BL/6.IL-23R$^{-/-}$ (hereafter referred as IL-23R$^{-/-}$) were obtained from Dr. Fiona Powrie (University of Oxford, Oxford, UK) with permission from Dr. Mohamed Oukka (University of Washington, Seattle, USA). Mice were bred and maintained in the specific pathogen-free animal facilities of Instituto de Medicina Molecular (Lisbon, Portugal). *Il17a$^{Cre}$R26R$^{eYFP}$* (referred to as IL-17 fate-mapping reporter) mice were bred in the MRC National Institute for Medical Research (Mill Hill, London, UK) animal facility under specified pathogen-free conditions. All experiments involving animals were done in compliance with the relevant laws and institutional guidelines and were approved by local and European ethics committees.

### Bone marrow chimeras

TCRδ$^{-/-}$ mice were lethally irradiated (950 rad), and the next day injected intravenously with a total of 5–10 × 10$^6$ whole bone marrow cells from Thy1.1$^+$ donor mice. For mixed BMCs, a total of 10$^7$ whole bone marrow cells of mixed (following a 1:1 ratio) Thy1.1$^+$ and IL-23R$^{-/-}$ (Thy1.2$^+$) origin, from age-matched animals, were injected in previously lethally irradiated TCRδ$^{-/-}$ hosts. In some experiments, BMCs supplemented with neonatal thymocytes were generated as previously described [17]. In brief, lethally irradiated TCRδ$^{-/-}$ hosts (Thy1.2$^+$) were injected, after 6 h, with neonatal thymocytes from pups (Thy1.1$^+$Thy1.2$^+$) within 48 h of birth. After 24 h, the host received 5–10 × 10$^6$ bone marrow cells from C57BL/6J (Thy1.1$^+$) donors. All BMCs were kept on antibiotics-containing water (2% Bactrim; Roche) for the first

4 weeks post-irradiation. The hematopoietic compartment was allowed to reconstitute for 8 weeks before the animals were used for experiments.

### EAE induction and scoring

BMCs were immunized s.c. in both flanks with 125 μg of myelin oligodendrocyte glycoprotein (MOG) 35–55 peptide (MEVG-WYRSPFSRVVHLYRNGK) (Eurogentec S.A.) emulsified in CFA solution (4 mg/ml of heat-inactivated *M. tuberculosis* in IFA) (Difco Laboratories). On the day of immunization and 2 days after, mice received 200 ng pertussis toxin (PTx) (List Biological Laboratories) in 100 μl PBS i.v. Mice were checked daily and scored for EAE clinical signs as described elsewhere [53]. In brief, the score system ranged from 0 to 5, with 0.5 increments, being score attributed to animals with no clinical signs of EAE and five representative of death. Score 1 consisted in limp tail; score 2 consisted in limp tail together with hind legs weakness; score 3 consisted in partial limb paralysis; and finally, score 4 consisted in complete hind leg paralysis.

### Administration of IFA, CFA, and TLR agonists *in vivo*

BMCs were injected s.c. in both flanks with CFA (4 mg/ml of heat-inactivated *M. tuberculosis* in IFA) or IFA (Difco Laboratories) emulsified in a 1:1 ratio (vol/vol) with sterile PBS; additionally, BMCs were given 200 ng of PTx i.v. on days 0 and 2 p.i. for additional adjuvant effect. On day 7 p.i., these BMCs were sacrificed and the spleen and dLN were harvested. For the individual TLR agonists (Pam$_3$CSK$_4$, Poly(I:C), LPS, and CpG), mice were injected s.c. in both flanks with 50 μg of each (all InvivoGen). BMCs were sacrificed on day 3 p.i., and spleen and dLN were harvested.

### Monoclonal antibodies

The following anti-mouse fluorescently labeled monoclonal antibodies (mAbs) were used (antigens and clones): CD3 (145.2C11, 17A2), TCRδ (GL3), Vγ1 (2.11), Vγ4 (UC3-10A6), CD44 (IM7), CD90.1 (Thy1.1; OX7), CD90.2 (Thy1.2; 53-2.1), IL-1RI (CD121a; JAMA-147), Ki67 (16A8), RORγ-t (Q31-378), T-bet (eBio4B10), IFN-γ (XMG1.2), and IL-17A (TC11.18H10.1). Antibodies were purchased from BD Biosciences, eBiosciences, or BioLegend.

### Cell preparation, flow cytometry, cell sorting, and analysis

For cell surface staining, single-cell suspensions were incubated in the presence of anti-CD16/CD32 (eBioscience) with saturating concentrations of combinations of the mAbs listed above. For the preparation of brain, spinal cord, lungs, and lamina propria cells, mice were perfused through the left cardiac ventricle with cold PBS. Lungs, spinal cord, and brain were dissected, and tissue was cut into pieces, and digested with collagenase type IV (0.5 mg/ml; Roche) and DNase I (0.10 mg/ml) (Sigma-Aldrich) in RPMI 1640 containing 5% fetal bovine serum (FBS) at 37°C for 30 min. For lamina propria cell preparation, small intestines were dissected, washed in ice-cold PBS, and Peyer's patches were excised. The organ was then cut into pieces and incubated with EDTA 0.05 M at 37°C for 20 min; cells were then washed and passed through a 100-μm cell strainer and then digested as the other organs. Mononuclear

cells were isolated by passing the tissue through a 40-μm cell strainer, followed by a 33% Percoll (Sigma-Aldrich) gradient and 30-min centrifugation at 1,160 *g*. Mononuclear cells were recovered from the pellet, resuspended, and used for further analysis.

For intracellular cytokine staining, cells were stimulated with PMA (phorbol 12-myristate 13-acetate) (50 ng/ml) and ionomycin (1 μg/ml), in the presence of Brefeldin A (10 μg/ml) (all from Sigma) for 3 h at 37°C. Cells were stained for the identified above cell surface markers, fixed 30 min at 4°C and permeabilized with the Foxp3/Transcription Factor Staining Buffer set (eBioscience) in the presence of anti-CD16/CD32 (eBioscience) for 10 min at 4°C, and finally incubated for 1 h at room temperature with identified above cytokine-specific Abs in permeabilization buffer. Cells were analyzed using FACSFortessa (BD Biosciences) and FlowJo software (Tree Star).

For cell sorting, peripheral lymph nodes (pLN) were prepared and stained for cell surface markers as mentioned above and then electronically sorted on a FACSAria (BD Biosciences).

### *In vitro* γδ T-cell stimulation

CD3$^+$TCRδ$^+$eYFP$^-$ cells were FACS-sorted from the pLN of IL-17 fate-mapping reporter mice and cultured *in vitro* at the concentration of 2–3 × 10$^4$ cells per well for 72 h in the presence of combinations of IL-1β (10 ng/ml), IL-23 (10 ng/ml), IL-6 (10 ng/ml), TGF-β (10 ng/ml), and plate-bound anti-CD3 mAb (10 μg/ml). Homeostatic IL-7 and IL-21 (10 ng/ml, each) were also added. All cytokines were from PeproTech, except TGF-β and IL-23, which were from R&D Systems.

### Statistical analysis

The statistical significance of differences between populations was assessed with the Kruskal-Wallis test (nonparametric one-way ANOVA) or by using a two-tailed nonparametric Mann–Whitney *U*-test, when applicable. The $P$-values < 0.05 were considered significant and are indicated on the figures.

**Expanded View** for this article is available online.

### Acknowledgements

We thank the precious assistance of the staff of the Flow Cytometry and Rodent facilities of iMM Lisboa, IGC, and The Francis Crick Institute; and Afonso Almeida, Karine Serre, Margarida Sanches-Vaz, Miguel Muñoz-Ruiz, Paula Vargas Romero and Rita Domingues (iMM Lisboa), Birte Blankenhaus (IGC), and Sara Omenetti (Crick) for helpful discussions and technical advice. We are also grateful to Mohammed Oukka (University of Washington, USA) and Fiona Powrie (Oxford University, UK) for provision of IL-23R$^{-/-}$ mice; and Shizuo Akira (Osaka University, Japan) for MyD88$^{-/-}$ mice. This work was funded by the European Research Council (CoG_646701 to B.S.-S.), Horizon 2020 (TwinnToInfect; grant agreement no. 692022), Wellcome Advanced Investigator Grant (to B.S.), and Fundação para a Ciência e Tecnologia (PD/BD/105855/2014 to P.H.P.; IF/00013/2014 to J.C.R.).

### Author contributions

PHP designed and performed experiments, analyzed the data, and wrote the manuscript; NG-S, NS, AI, and SM assisted in the experiments; BS assisted in the experimental design and provided key research tools; JCR provided

technical supervision and assisted in the experimental design; BS-S supervised the research and wrote the manuscript.

## Conflict of interest

The authors declare that they have no conflict of interest.

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
