## [Review Process File · EMBO Reports]

Manuscript EMBO-2017-44200

IL-23 drives differentiation of peripheral $\gamma\delta$ 17 T cells from adult bone marrow-derived precursors

Pedro H. Papotto, Natacha Gonçalves-Sousa, Nina Schmolka, Andrea Iseppon, Sofia Mensurado, Brigitta Stockinger, Julie C. Ribot, and Bruno Silva-Santos

Corresponding author: Bruno Silva-Santos, Instituto de Medicina Molecular

Review timeline:	Submission date:	10 March 2017
	Editorial Decision:	26 April 2017
	Revision received:	21 July 2017
	Editorial Decision:	25 July 2017
	Revision received:	28 July 2017
	Accepted:	31 July 2017

Editor: Achim Breiling

Transaction Report:

1st Editorial Decision

26 April 2017

Thank you for the submission of your research manuscript to EMBO reports. We have received the reports from the two referees that were asked to evaluate your study, which can be found at the end of this email.

As you will see, both referees highlight the potential interest of the findings. However, they have raised a number of concerns and suggestions to improve the manuscript, or to strengthen the data and the conclusions drawn. As the reports are below, I will not detail them here, but it will be important to improve the statistical analysis as pointed out by referee #1 and to address all the concerns of referee #2.

Given the constructive referee comments, we would like to invite you to revise your manuscript with the understanding that all referee concerns must be addressed in the revised manuscript and in a point-by-point response. Acceptance of your manuscript will depend on a positive outcome of a second round of review. It is EMBO reports policy to allow a single round of revision only and acceptance or rejection of the manuscript will therefore depend on the completeness of your responses included in the next, final version of the manuscript.

REFeree REPORTS

Referee #1:

The study by Papotto and colleagues addresses the relevance of peripheral differentiation of gd17 T cells in experimental autoimmune encephalomyelitis. This is a well-written and well-designed study convincingly demonstrating that, contrary to the widely held believe that gd17 cells are pre-programmed in the fetal thymus, IL-17 can be induced in adult gd T cells in the periphery.

Major comments:

(1) The statistical analysis of the present study deserves closer attention. Throughout the manuscript group sizes are defined as n=2-7 without explanation why group sizes vary so dramatically. It is unclear how appropriate statistical tests could be applied to groups consisting of only 2 animals. The meaning of independent and representative experiments is unclear, and figure legends should be more explicit: were data in the figures compiled and pooled from two independent experiments, or are the data shown from one out of two independent experiments?

(2) No details are given how datasets were tested for normal distribution to be able to apply t-tests. How were groups of n=2 treated? Comparisons in Figs. 1D, 1G, 2E, 3K, 4D and 4F in fact require ANOVA-based tests.

Referee #2:

This study on the development of IL-17-secreting $\gamma\delta$ in an autoimmune disease model extends previous publications on the pathogenic role of these cells in EAE. The findings are interesting and novel and the experiments appear to have been carefully performed.

Specific comments:

1. As I understand the experimental design, TCR-delta KO mice are irradiated and then given bone marrow cells from wild-type Thy1.1+ mice and in the experiments shown in figures 1 and 2, these were compared (for EAE and T cell development) with wildtype mice that had not been irradiated or given any cell transfer. Is this a valid comparison? Would irradiated WT mice given WT bone marrow not be the proper control? We have no way of know how the irradiation and BM transplant affected T cell development and induction of EAE, so how can this be distinguished from the role of gamma delta T cells?

2. For intracellular cytokine staining, the cells have been stimulated with PMA and ionomycin in the presence of brefaldin-a (BFA). This can artificially elevate the real frequency of cytokine secreting T cells. Normally IL-17-secretion by gamma delta T cells during EAE can be detected in LN ex vivo by incubation with BFA alone. The data in Fig 1B show that gamma delta T cells from spleen and LN of naive mice secrete IFN-gamma but not IL-17. This is surprising as up to 20% of gamma delta T cells from LN of naïve mice secrete IL-17 without stimulation ex vivo.

3. Fig 2: The authors have looked in the LN, spleen and thymus for IL-17 secreting gamma delta T cells an only find them in the LN early in development of EAE. The lung and gut have both been implicated as sites for development of pathogenic T cells in EAE. It would be interesting to look in the lungs and gut.

4. The data in Fig 2D and E is interesting but the role of CFA is unclear because of the co-injection of PT, which is known to multiple effects outside of adjuvant activity. This might have been cleaner if done with CFA alone.

5. The lack of a role for IL-1 in promoting IL-17-secreting gamma-delta T cells based data in Fig 2 F and G is surprising given the previous reports on the crucial role of IL-1 with IL-23 in innate IL-17 production by gamma delta T cells. MYD88-independnat IL-1 signalling has been reported, so the authors need to use IL-1R1 KO mice or constrain their conclusions on IL-1 or confine them to TLRs.

We thank the Reviewers for their constructive criticism of our manuscript, which prompted additional experiments and clarifications that improved the quality of our (revised) paper. We provide below a point-by-point reply to the concerns that were raised.

Referee #1:

(1) The statistical analysis of the present study deserves closer attention. Throughout the manuscript group sizes are defined as n=2-7 without explanation why group sizes vary so dramatically. It is unclear how appropriate statistical tests could be applied to groups consisting of only 2 animals. The meaning of independent and representative experiments is unclear, and figure legends should be more explicit: were data in the figures compiled and pooled from two independent experiments, or are the data shown from one out of two independent experiments?

These discrepancies were due to our prioritization of the immunized groups, due to their intra-group variation and because it is well established that bone marrow chimeras lack gd17 T cells (Haas, 2012). However, throughout the paper it is clear that all naïve BMCs behave similarly, showing consistency in the phenotype previously described. Moreover, the only figure we had groups with n= 2 was Fig 1A-D, for which no statistic test was applied. This notwithstanding, we have performed new experiments in order to increase the small group sizes and overcome this valid concern of Referee #1. Sample sizes can now be found in each figure, for each experiment. We still have a variation of group sizes, in the case of experiments using naïve BMCs, as in order to reduce the number of mice being used in this study we have continued to prioritize the immunized groups over the controls essentially and consistently devoid of gd17 T cells. Also, we have now carefully explained the independent repetition of experiments in the figure legends.

(2) No details are given how datasets were tested for normal distribution to be able to apply t-tests. How were groups of n=2 treated? Comparisons in Figs. 1D, 1G, 2E, 3K, 4D and 4F in fact require ANOVA-based tests.

We apologize for this shortcoming and fully acknowledge that, due to the sample size constraints of our bone marrow chimeras, the datasets do not comply with normal distributions. Thus, we have now performed non-parametric tests (Mann-Whitney U or Kruskal-Wallis) in all our comparisons.

Of note, no statistical test had been applied to the group with n= 2 at the time of first submission; however, have now performed additional experiments to increase sample size, as discussed in (1), so that is no group with n= 2 in the revised manuscript.

Moreover, we fully agree that figures 1G, 2D and 2F (formerly 2E) should be analyzed using ANOVA-based tests and thus we performed a non-parametric one-way ANOVA (Kruskal-Wallis test with Dunn's multiple comparisons test). However, for figures 1D, 3K (now Supplementary Figure 3), 4D, 4F and 4H, we have applied the Mann-Whitney test between the naïve and immunized samples (or NeoThy and BM, in the case of figure 4H) within each organ, as they are not meant to be cross-compared, and were depicted together only to facilitate the navigation through the data.

Referee #2:

(1) As I understand the experimental design, TCR-delta KO mice are irradiated and then given bone marrow cells from wild-type Thy1.1+ mice and in the experiments shown in figures 1 and 2, these were compared (for EAE and T cell development) with wildtype mice that had not been irradiated or given any cell transfer. Is this a valid comparison? Would irradiated WT mice given WT bone marrow not be the proper control? We have no way of know how the irradiation and BM transplant affected T cell development and induction of EAE, so how can this be distinguished from the role of gamma delta T cells?

Although we show in Fig 1C the comparison between Thy1.1 : TCRd^{-/-} BMC and C57Bl/6 animals regarding the clinical course of EAE, in all the following panels/ figures the “naïve” group refers to non-immunized Thy1.1 : TCRd^{-/-} BMC. We have chosen to depict C57Bl/6 mice in Fig 1C simply as a way to validate EAE in BMC and show that both groups similarly develop the disease. In order to avoid this misinterpretation, we have changed the text and the figure legends.

(2) For intracellular cytokine staining, the cells have been stimulated with PMA and ionomycin in the presence of brefaldin-a (BFA). This can artificially elevate the real frequency of cytokine secreting T cells. Normally IL-17-secretion by gamma delta T cells during EAE can be detected in LN ex vivo by incubation with BFA alone. The data in Fig 1B show that gamma delta T cells from spleen and LN of naïve mice secrete IFN-gamma but not IL-17. This is surprising as up to 20% of gamma delta T cells from LN of naïve mice secrete IL-17 without stimulation ex vivo.

We believe that this comment arose from the misunderstanding in the first question above. If our naïve group was composed by wild-type C57Bl/6, it would, indeed, be surprising that no IL-17⁺ gd T cells were observed after PMA + Ionomycin treatment. However, the “naïve” group corresponds to unimmunized Th1.1 : TCRd^{-/-} BMC, and it is expected (Haas et al. Immunity 2012) that we do not observe IL-17⁺ gd T cells in this setting, even after PMA + ionomycin treatment. Moreover, although we fully agree with the comment on BFA, in our case, as our starting point is the complete absence of gd17 cells, we believe that the observed gd17 cells were generated due to the inflammatory stimulus, and PMA+ ionomycin restimulation is thus suitable to unravel the *de novo* differentiated gd17 T cells.

(3) Fig 2: The authors have looked in the LN, spleen and thymus for IL-17 secreting gamma delta T cells and only find them in the LN early in development of EAE. The lung and gut have both been implicated as sites for development of pathogenic T cells in EAE. It would be interesting to look in the lungs and gut.

We have now immunized BMCs and sacrificed them before the onset of EAE (d7 p.i.), and as suggested by referee #2, we analyzed the lungs and lamina propria of these mice, together with different lymphoid organs. We did not observe gd17 T cells neither in the small intestine lamina propria nor in the mesenteric lymph nodes. However, in the lungs, like in cervical lymph nodes, we observed a small population (<3%) of gd17 T cells (please see figure below). This can indicate that these cells are also being generated in these other organs, but also that these cells are already recirculating to become licensed to enter the central nervous system (Odoardi, 2012). In any case, given the fact that the draining lymph nodes account for the major population of gd17 T cells in this initial phase of EAE (see figure below), we believe that their generation is mainly occurring in the dLN.

(4) The data in Fig 2D and E is interesting but the role of CFA is unclear because of the co-injection of PT, which is known to have multiple effects outside of adjuvant activity. This might have been cleaner if done with CFA alone.

We thank the Reviewer for this suggestion. We have now performed this experiment, and found that without PTx, CFA is in fact incapable of inducing substantial gd17 T cell differentiation (please see figure below). The low frequencies of gd17 T cells observed upon CFA alone are similar to IFA + PTx. We believe that PTx is required to maximize the innate stimuli provided by CFA (but not IFA) through the production of innate cytokines, as previously described (Fedele, 2005; Ronchi, 2015). We discuss this idea on page 10 of the revised manuscript.

(5) The lack of a role for IL-1 in promoting IL-17-secreting gamma-delta T cells based data in Fig 2 F and G is surprising given the previous reports on the crucial role of IL-1 with IL-23 in innate IL-17 production by gamma delta T cells. MYD88-independent IL-1 signalling has been reported, so the authors need to use IL-1R1 KO mice or constrain their conclusions on IL-1 or confine them to TLRs.

We fully agree with the Reviewer's comment and apologize for over-interpreting of our previous data. To address this issue, we generated mixed BMC using IL-1R1-deficient (or sufficient) cells and immunized them subcutaneously with CFA+PTx. As shown below (panel J), IL-1R1-deficient cells were capable of generating gd17 T cells, similarly to their WT counterparts. This excludes the relevance of MYD88-independent IL-1R1 signaling in this process. Hence, our results demonstrate that *de novo* generation of gd17 T cells *in vivo* is strongly dependent on IL-23 (Fig 3G) but not IL-1 (below and new Fig 3J in the revised manuscript) signals.

3rd Editorial Decision

25 July 2017

Thank you for the submission of your revised manuscript to our editorial offices. We have now received the reports from the referees that were asked to re-evaluate your study (you will find enclosed below). As you will see, both referees now support the publication of your manuscript in EMBO reports.

Before we can proceed with formal acceptance, I have the following editorial requests that need to be addressed in a final revised version:

For a Short Report, results and discussion must be combined into one section (Results and Discussion). Please do that for your manuscript text. Also, please remove the significance statement, a feature an EMBO reports paper usually does not have. I suggest moving this, or parts of it, to the end of the introduction (to provide a brief overview of the work and a summary).

The title is presently too long. Please provide a title with no more than 100 characters including spaces. Then, please also shorten the abstract to below 176 words and provide it written in present tense.

Please add up to 5 keywords to the title page, and a conflict of interest statement before the acknowledgements.

The font size in many of the graphs is rather small and hard to read at 100%. Please use bigger fonts and refer to our guidelines for figure preparation.

For a short report we allow up to 5 main and EV figures. Thus, in case you can also split the data and have fifth figures, in order to increase legibility.

REFEREE REPORTS

Referee #1:

The manuscript is suitable for publication in EMBO reports without revision.

Referee #2:

The authors have done an excellent job addressing my comments. I have no further concerns.

2nd Revision - authors' response

28 July 2017

The authors made the requested changes and submitted the final version of their manuscript.

4th Editorial Decision

31 July 2017

I am very pleased to accept your manuscript for publication in the next available issue of EMBO reports. Thank you for your contribution to our journal.

YOU MUST COMPLETE ALL CELLS WITH A PINK BACKGROUND

Corresponding Author Name: Bruno Silva-Santos

Manuscript Number: EMBOR-2017-44200